# Dose–Response Associations Between Diet and Risk of Rheumatoid Arthritis: A Meta-Analysis of Prospective Cohort Studies

**DOI:** 10.3390/nu16234050

**Published:** 2024-11-26

**Authors:** Yuanyuan Dong, Darren C. Greenwood, James Webster, Chinwe Uzokwe, Jinhui Tao, Laura J. Hardie, Janet E. Cade

**Affiliations:** 1Nutritional Epidemiology Group, School of Food Science and Nutrition, University of Leeds, Leeds LS2 9JT, UK; fscau@leeds.ac.uk (C.U.); j.e.cade@leeds.ac.uk (J.E.C.); 2Leeds Institute for Data Analytics, Faculty of Medicine and Health, University of Leeds, Leeds LS2 9JT, UK; d.c.greenwood@leeds.ac.uk; 3Applied Health Research Unit, Nuffield Department of Population Health, University of Oxford, Oxford OX3 7LF, UK; james.webster@ndph.ox.ac.uk; 4Department of Rheumatology and Immunology, The First Affiliated Hospital of University of Science and Technology of China, Hefei 230001, China; taojinhui@ustc.edu.cn; 5Division of Clinical and Population Sciences, Leeds Institute of Cardiovascular and Metabolic Medicine, School of Medicine, University of Leeds, Leeds LS2 9JT, UK; l.j.hardie@leeds.ac.uk

**Keywords:** rheumatoid arthritis, diet, meta-analysis, dose–response, nutrient intake

## Abstract

To provide a systematic and quantitative summary of dietary factors and rheumatoid arthritis (RA) risk. A systematic review and meta-analysis included prospective cohort studies from 2000 to 2024 reporting relative risks (RRs) with 95% confidence intervals (CIs) for RA incidence relating to 32 different dietary exposures. Linear and non-linear dose–response analyses were conducted. Thirty studies were included, involving 2,986,747 participants with 9,677 RA cases. Linear dose–response analysis suggested that each 2-unit per week increase in total alcohol intake was linked to 4% risk reduction (RR (95%-CI), heterogeneity (*I*^2^), NutriGrade score: 0.96 (0.94, 0.98), 58%, moderate certainty), and beer consumption was associated with a 10% reduction per 2 units/week increase (0.90 (0.84, 0.97), 0%, very low certainty). Each 2-unit/week increase in total alcohol intake was associated with a 3% decrease in seropositive RA risk (0.97 (0.96, 0.99), 28%, moderate certainty). Increased intakes of fruit (per 80 g/day) and cereals (per 30 g/day) were associated with 5% (0.95 (0.92, 0.99), 57%, moderate certainty) and 3% (0.97 (0.96, 0.99), 20%, moderate certainty) reduced risk, respectively. Conversely, tea consumption showed a 4% increased risk per additional cup/day (1.04 (1.02, 1.05), 0%, moderate certainty). Non-linear associations were observed for total coffee, vegetables, oily fish, and vitamin D supplementation. Data on dietary patterns and specific micronutrients were limited. The findings suggest that moderate alcohol consumption and a higher intake of fruits, oily fish, and cereals are associated with a reduced risk of RA, while tea and coffee may be linked to an increased risk. Optimising dietary intake of certain food components may reduce RA risk, despite moderate-quality evidence.

## 1. Introduction

Rheumatoid arthritis (RA) is the most common systemic autoimmune disease, characterised by the presence of autoantibodies, like rheumatoid factor (RF) and anti-citrullinated protein antibody (ACPA) [1]. While it can lead to long-term disability and increased morbidity [2], early diagnosis and guideline-based treatment significantly mitigate these effects. The Global Burden of Disease (GBD) 2021 study stated that the incidence of RA was set to continue increasing, with an estimated global prevalence of 17.6 million cases in 2020 and notable regional variations [3]. Only a few established risk elements for RA are recognised, including older age, female sex, infections, air pollution, and cigarette smoking [4].

Diet has been associated with other autoimmune diseases [5], and both genetic and environmental factors can influence the development of RA [6]. A better understanding of how diet affects RA could help inform more effective prevention strategies. Several dietary elements, particularly plant-based foods rich in fibre, vitamins, and antioxidants, can reduce systemic inflammation by lowering levels of *C*-reactive protein (CRP), interleukin-6 (IL-6), and tumour necrosis factor-α (TNF-α) [7]. Other components, such as short-chain fatty acids and immune-enhancing nutrients like vitamin D [8] and folate [9], may contribute to inflammation regulation. 

Nevertheless, the roles of dietary patterns and individual components remain unclear. For instance, the Mediterranean diet (MD) has been associated with lower inflammatory markers and reduced risks of chronic diseases like type 2 diabetes and cardiovascular disease [10,11], but direct evidence for its effect on RA risk is still limited. The benefit of alcohol consumption for RA risk has been the subject of previous studies [12]. A 2014 meta-analysis demonstrated a non-linear trend indicating that low-to-moderate alcohol intake reduced the risk of RA [13], but it included non-peer-reviewed literature. In addition, data on specific alcoholic beverages are scarce. Although several case–control studies have suggested that tea consumption may reduce RA disease activity [14,15], findings from the UK Biobank indicated that it could increase RA risk [16]. The French Society for Rheumatology (FSR) recommends omega-3 supplementation for inflammatory rheumatic diseases [17]; however, a recent meta-analysis of randomised controlled trials (RCTs) found no reduction in RA symptoms or inflammation [18]. There is also evidence regarding the impact of anti-rheumatic therapies on periodontal health, highlighting the complex relationship between lifestyle factors and RA progression [19]. The inconsistency in these findings points to the need for further research to clarify the effects of food groups and nutrients on RA risk and assess the credibility of the evidence.

Two recent systematic reviews examined the role of diet in preventing RA, but did not include individual dietary components in their search strategies [20,21]. To address these gaps, this study aims to synthesise the existing data and provide high-quality evidence on the associations of dietary components with the risk of RA in a dose–response analysis.

## 2. Materials and Methods

The findings from this systematic review and dose–response meta-analysis are reported based on the Preferred Reporting Items for Systematic review and Meta-Analysis (PRISMA) guidelines [22]. The protocol for this review was registered on PROSPERO as CRD42022320959. 

### 2.1. Search Strategy

We conducted a systematic search of all articles published from 1 January 2000 to 30 April 2024 using databases, including Medline, Embase, Web of Science, and the Cochrane Library. In brief, we searched for prospective cohort studies that reported associations of dietary factors (food groups, beverages, food components, nutrients, and dietary patterns) with RA incidence. Full details of the search strategy are provided in Appendix A. The search followed the PICO framework [23], as shown in Appendix A. To avoid missing any publications, the references of relevant papers were checked. 

### 2.2. Inclusion Criteria and Study Selection

The eligibility criteria for inclusion in the meta-analysis were as follows: (1) original research studies; (2) human studies conducted on adults; (3) prospective studies that provided data on dietary consumption factors; (4) studies reporting the effect sizes, including hazard ratio (HR) and relative ratio (RR) with 95% confidence intervals (CIs), where the outcome was RA incidence; (5) English-language studies; and (6) studies for which full texts were available. The exclusion criteria were: (1) reviews and book chapters, or secondary research evidence such as meta-analyses; (2) animal studies, letters, comments, cross-sectional studies, and ecological studies; (3) overlapping studies or data; (4) studies without dietary exposure or measurement (e.g., studies that only report smoking and RA risk); (5) studies focusing on RA-related symptoms, life quality, functional status, or disease activity; and (6) Mendelian randomisation studies.

The publications were independently assessed by YD, JC, DG, LH, and CU, and any discrepancies were resolved through discussion with the senior authors (YD and JC). After deduplication, articles were screened for eligibility based on their titles and abstracts, followed by a detailed assessment using the Systematic Review Accelerator [24]. If there were duplicate publications from the same cohort, we included the study with the longer follow-up or the larger number of participants or incident cases, provided the data were sufficient to generate effect estimates for meta-analysis. 

Our research was based on the 1987 American Rheumatism Association (ACR) classification criteria for RA [25], expecting that prospective cohort studies would be incorporated into the research.

### 2.3. Data Extraction and Quality Assessment

JW and YD extracted data from each eligible article using a predesigned form, including the study details (first author, cohort name, publication year, study location, follow-up duration, dietary exposure, cohort size, and the number of RA cases), participant characteristics (age range, health status at entry, and sex), assessment methods (mean/median or range of food/nutrient intake, and RA case ascertainment), statistical analysis (models and confounders), and effect sizes (incidence rate ratio [IRR], HR, and RR) from the most fully adjusted models. A meta-analysis was conducted only for exposure–outcome associations with data from at least two studies. For the dose–response meta-analysis, studies were excluded if they lacked data on non-cases or person-years. 

The study quality was assessed using the Newcastle–Ottawa Scale (NOS), a tool commonly employed for evaluating observational studies based on their selection, comparability, and outcome assessment [26]. Studies scoring above six were considered high-quality, while those scoring six or below were considered low-quality. We used the NutriGrade tool (developed in Nuthetal, Brandenburg, Germany) to evaluate the certainty of the meta-evidence for each dietary exposure’s association with RA [27]. NutriGrade assigns a score (out of 10) by considering factors including study design, risk of bias, precision, heterogeneity, directness, publication bias, funding bias, effect size, and dose–response. The meta-evidence was classified as high (8~10), moderate (6~7.99), low (4~5.99), or very low (0~3.99). Moderate-quality evidence indicates robust findings with some limitations, while low-quality evidence reflects greater uncertainty.

### 2.4. Statistical Methods

Associations for both highest- versus lowest-category comparisons and dose–response analyses were investigated. RR or HR estimates from individual studies were combined using a random effects model. The methods described by Greenland and Longnecker [28] and Orsini et al. [29] were used for the dose–response meta-analysis. To estimate dose amounts, the mean, median, or midpoint values of the upper and lower limits were used. When the ranges were open-ended, we estimated the limits based on adjacent intervals. Restricted cubic splines with three knots (at 10%, 50%, and 90% of exposure distribution) to handle potential non-linear relationships were used. For the non-linear dose–response analysis, we included studies that reported the RR with 95% CIs for at least three categories of exposure in order to estimate a dose–response trend. 

To standardise the dietary exposures across studies, we converted the consumption data into a common scale, such as grams per week for alcoholic beverages. The standard conversions are shown in Appendix A. To ensure consistency in the analysis, we used the lowest exposure as the reference category. If the reference group was different from the lowest category, we recalculated the effect size using the method of Orsini et al. [30]. Meta-analyses were performed for food groups, beverages (e.g., total alcohol, wine, beer, coffee types, tea, fruits, vegetables, meats, fish, dairy products, cereals), nutrients, and phytochemicals (e.g., vitamin C, D, A, E, *n*-3 polyunsaturated fatty acids, PUFAs, carotenoids). 

Heterogeneity across studies was assessed by *I*^2^ statistic, which quantifies the percentage of total variation due to heterogeneity rather than chance, with *I*^2^ > 50% indicating substantial heterogeneity. When substantial heterogeneity was detected, we conducted subgroup analyses for dietary factors with data from at least 5 studies, categorised by study characteristics such as age (≥50 vs. <50), gender (both sexes vs. female only), location (Europe, America, Asia), follow-up duration (<10 years vs. ≥10 years), dietary assessment method (food frequency questionnaires, FFQs vs. food diaries), case ascertainment method (self-reported vs. registry-based data), and number of RA cases (<500 vs. ≥500). 

The potential effects of small-study biases, including publication bias, were evaluated using funnel plots, Egger’s test [31], and Begg’s test [32], provided there were a sufficient number of studies to perform the test. Sensitivity analyses were conducted to evaluate the robustness of the results by excluding 1 study at a time when more than 3 studies were available. 

All the statistical analyses were performed using R version 4.2.0 (R Foundation for Statistical Computing, Vienna, Austria). A two-tailed *p*-value of <0.05 was regarded as statistically significant. 

## 3. Results

### 3.1. Literature Search

The search strategy identified 3525 unique citations, of which 55 full-text articles were assessed in detail (Figure 1). The reasons for exclusion during full-text screening are provided in Appendix A. 

### 3.2. Characteristics of Included Studies

The characteristics of the 30 studies included in the analysis, which investigated the relationship between dietary factors and the risk of developing RA [12,16,33,34,35,36,37,38,39,40,41,42,43,44,45,46,47,48,49,50,51,52,53,54,55,56,57,58], are summarised in Figure 2 and Appendix A. Of these, 22 articles focused exclusively on women [12,33,34,35,36,38,39,40,41,42,43,44,45,46,47,54,55,56,57,58,59,60], and 8 included both sexes [16,37,48,49,50,51,52,53]. The studies were primarily nationally representative: 12 were conducted in Europe [16,36,37,38,41,42,43,48,49,50,52,54], 15 in America [12,33,34,35,39,40,44,45,46,47,55,56,57,58,59], and 3 in Asia [50,51,52,53,58,60].

Overall, during the follow-up periods ranging from 4 to 30 years, 14 prospective studies included 9677 RA cases and 8772 seropositive RA cases among 2,986,747 participants with an age range spanning from 20 to 98 years (Appendix A). Key covariates included smoking (n = 29), age (n = 26), body mass index (BMI) (n = 22), total energy intake (n = 15), physical activity (n = 13), and alcohol consumption (n = 12).

Ten prospective studies were included in the meta-analysis for the consumption of alcohol (nine reports) [16,33,34,37,41,48,51,54,57], five studies for wine (four reports) [12,41,54,57], five for beer (four reports) [12,41,54,57], five for liquor (four reports) [12,41,54,57], six for total coffee [16,46,47,52,54,55], four for caffeinated coffee [46,47,54,55], five for decaffeinated coffee [16,46,47,54,55], five for tea [16,46,47,54,55], three for caffeine [46,47,55], three for sugar-sweetened soda (two reports) [54,59], seven for fruits (six reports) [16,33,34,35,38,49], seven for vegetables (six reports) [16,33,34,35,38,49], four for total meat [36,42,50,58], three for red meat [33,42,58], two for processed meat [16,42], three for poultry [16,42,58], five for total fish (four reports) [36,43,49,56], three for oily fish [16,36,49], two for legumes [35,36], three for total dairy [36,42,45], two for milk [42,45], two for cheese [16,42], four for cereal products (three reports) [16,34,36], four for *n*-3 PUFAs (three reports) [33,34,43], four for vitamin C (three reports) [35,40,49], four for vitamin D (three reports) [39,45,49], three for vitamin D supplements (two reports) [39,45], three for vitamin A (two reports) [40,49], four for vitamin E (three reports) [35,40,49], and three for carotenoids (two reports) [35,40] (Appendix A).

### 3.3. Alcohol Consumption and RA Risk

An inverse association was observed between total alcohol consumption and RA risk across eight studies (RR: 0.76, 95% CI: 0.68, 0.85, *I*^2^ = 56%, P-heterogeneity = 0.03; Figure 3) when comparing the highest and lowest categories. This inverse relationship was seen with a total alcohol intake of up to 60 g/week (P-non-linearity = 0.004, Figure 4). Similarly, a non-linear dose–response association for beer was detected (P-non-linearity = 0.013, Appendix A). Linear analysis showed a 4% risk reduction for each additional 2 units of total alcohol intake (RR: 0.96; 95% CI: 0.94, 0.98, *I*^2^ = 58%, P-heterogeneity = 0.01), and a 10% risk reduction for each additional 2 units of beer consumption (RR: 0.90; 95% CI: 0.84, 0.97, *I*^2^ = 0%, P-heterogeneity = 0.98; Appendix A). While there was some evidence of an association for each additional 2 units/week of wine consumption (RR: 0.98; 95% CI: 0.97, 1.00; Appendix A), this was borderline non-significant. No evidence of association was observed for liquor intake. The NutriGrade scores showed moderate-quality evidence for total alcohol, low-quality evidence for wine and liquor, and very-low-quality evidence for beer (Appendix A).

A pooled analysis of four studies on seropositive RA found an RR of 0.84 (95% CI: 0.73, 0.96; Appendix A) [12,53,54,60] for total alcohol when comparing extreme categories. Non-linear dose–response analysis supported this inverse association (P-non-linearity < 0.001; Appendix A). Each 2 unit/week increase in total alcohol intake was associated with a 3% reduction in the risk of seropositive RA (RR: 0.97; 95% CI: 0.96, 0.99; Appendix A). 

### 3.4. Non-Alcoholic Beverages and RA Risk

Tea and caffeinated coffee consumption were associated with increased RA risk (tea: RR: 1.15, 95% CI: 1.00, 1.33, *I*^2^ = 53%, P-heterogeneity = 0.07; caffeinated coffee: RR: 1.30, 95% CI: 1.09, 1.54, *I*^2^ = 0%, P-heterogeneity = 0.67) when comparing extreme categories, while total coffee, decaffeinated coffee, caffeine, and soda showed no evidence of association with RA risk (Appendix A). We found a positive trend for tea with a pooled RR of 1.04 per cup/d (95% CI: 1.02, 1.05, *I*^2^ = 0%, P-heterogeneity = 0.53) in the linear dose–response analysis, but no conclusive dose–response association was observed for other beverages (Appendix A). A positive association between total coffee consumption and RA risk was noted in the non-linear trend analysis (P-non-linearity = 0.015, Appendix A). Additionally, there was no evidence of associations for decaffeinated coffee, tea, and RF-positive RA (Appendix A). The NutriGrade evaluations showed moderate-quality evidence for tea, lower-quality evidence for caffeinated coffee and sugar-sweetened soda, and very-low-quality evidence for total coffee, decaffeinated coffee, and caffeine (Appendix A).

### 3.5. Fruit, Vegetables, and RA Risk

Higher fruit intake was associated with reduced RA risk (RR: 0.88, 95% CI: 0.79, 0.97, *I*^2^ = 50%, P-heterogeneity = 0.09; Appendix A), with evidence of a non-linear dose–response trend (P-non-linearity = 0.002, Figure 4). A linear analysis of six studies also indicated that an increase in fruit intake of 80 g/d increments was inversely associated with a reduced RA risk of 5% (95% CI: 0.92, 0.99, *I*^2^ = 57%, P-heterogeneity = 0.04; Appendix A). No evidence of association was found for vegetables (RR: 0.90, 95% CI: 0.76, 1.06, *I*^2^ = 81%, P-heterogeneity < 0.01; Appendix A), though there was evidence of a non-linear dose–response association (P-non-linearity = 0.023, Figure 4). The NutriGrade evaluations indicated moderate-quality evidence for fruits, and lower-quality evidence for vegetables (Appendix A).

### 3.6. Meat, Fish, and RA Risk

No evidence of RA risk associations emerged for total meat, red meat, processed meat, poultry, or total fish consumption (Appendix A). A non-linear inverse association between RA risk and oily fish intake was observed (P-non-linearity < 0.001; Figure 4). The NutriGrade assessments rated the evidence for total meat, processed meat, poultry, total fish, and oily fish as low-quality, and the evidence for red meat as very-low-quality (Appendix A).

### 3.7. Other Food Components and RA Risk

Higher cereal intake was associated with a reduced risk of RA (RR: 0.87, 95% CI: 0.80, 0.96, *I*^2^ = 0%, P-heterogeneity = 0.86; Appendix A) when comparing extreme categories. The dose–response analysis indicated a 3% risk reduction for each additional 30 g/d increase (RR: 0.97, 95% CI: 0.96, 0.99, *I*^2^ = 20%, P-heterogeneity = 0.29; Appendix A). However, there was no evidence of associations between legumes, dairy, or cheese and RA risk (Appendix A). The NutriGrade scores revealed moderate-quality evidence for cereal products, low-quality evidence for legumes, dairy, and milk, and very-low-quality evidence for cheese (Appendix A).

### 3.8. Nutrients and RA Risk

#### 3.8.1. Dietary Macronutrients

No evidence of an association between *n*-3 PUFA intake and RA risk was observed when comparing the highest with the lowest categories (Appendix A), although a borderline inverse non-linear association was noted (*P*-non-linearity = 0.063; Appendix A). A slight reduction in RA risk was seen for each additional 100 mg/d of n-3 PUFAs, but the evidence remains insufficient (RR: 0.96, 95% CI: 0.90, 1.03, *I^2^* = 74%, *P*-heterogeneity = 0.13; Appendix A). A meta-analysis for other macronutrients (e.g., energy, protein, carbohydrates, total fat) was not possible due to limited data, despite alcohol having previously been analysed. The NutriGrade scores revealed very-low-quality evidence for *n*-3 PUFAs (Appendix A).

#### 3.8.2. Dietary Micronutrients

Vitamin D supplement intake showed an inverse non-linear association with RA risk (*P*-non-linearity = 0.003, Appendix A). The risk of RA decreased by ~20% with an increase in vitamin D supplement intake of ≤~320 IU/d. No other evidence of association was found between dietary vitamins or carotenoids and RA risk (Appendix A). 

The NutriGrade evaluations rated the evidence quality as low for vitamins, β-cryptoxanthin, and lycopene, and very low for α-carotenoids, β-carotenoids, and lutein/zeaxanthin (Appendix A).

### 3.9. Subgroup Analysis and Bias Analysis

The stratified analyses showed evidence of heterogeneity between subgroups. An inverse association between total alcohol consumption and risk of RA was confirmed in most of the subgroup analyses, except for studies conducted in Asia (Appendix A). Significant associations with RA risk were observed for the consumption of decaffeinated coffee, fruit, and vegetables in studies involving women, and in those conducted in America, but no association was found in studies including both sexes or those conducted in other regions (Appendix A–g). The quality assessment, detailed in Appendix A, showed that 80% (24 out of 30) of the included studies were of good quality. Sensitivity analyses confirmed that excluding any single study did not change the pooled effect sizes (Appendix A). Although the funnel plots and Egger’s test indicated a publication bias for coffee intake (*p* < 0.01), no bias was detected for other foods (Appendix A). Trim-and-fill adjustments showed consistent results, suggesting a minimal impact from publication bias (Appendix A).

## 4. Discussion

This is the first comprehensive meta-analysis of the associations between RA risk in adults and the consumption of a range of foods, beverages, and nutrients, conducted to concurrently examine the potential protective and harmful effects of diet on RA risk in a dose-specific association. 

Our results identified associations between a higher intake of alcohol, fruits, and cereals and a lower risk of RA, while higher consumption of tea and caffeinated coffee was linked to an increased risk. Dose–response analyses further clarified these associations, revealing J-shaped trends for total alcohol and beer, protective linear relationships for fruit and cereals, and non-linear patterns for vegetables, oily fish, and vitamin D supplements. Notably, tea intake showed a positive linear association with RA risk, while total coffee consumption followed a non-linear trend. Evidence regarding dietary patterns and micronutrients was limited. Although based on low-to-moderate-quality meta-evidence, the findings provide new insights into the potential role of diet in RA risk, supporting their integration into prevention strategies.

To our knowledge, this is the first meta-analysis to focus on the associations of fruit, vegetables, and cereal intake with RA risk. Although the World Health Organization recommends consuming over 400 g (about five servings) of fruits and vegetables daily to prevent non-communicable diseases [61], evidence regarding the direct effect of fruit and vegetable intake on RA is rare. Our analysis, which included 7294 RA cases from Europe and America, found moderate-certainty evidence of an association between fruit intake and RA risk. This aligns with previous meta-analyses that linked fruit intake to reduced risks of inflammatory bowel disease, multiple sclerosis, and type 2 diabetes [62,63,64]. In addition, two RCTs have demonstrated the anti-inflammatory effects of fruit- and vegetable-rich diets compared to placebos [65,66]. A diet rich in fruits and vegetables is biologically plausible for enhancing immune function [67], reducing inflammation [68], and preventing autoimmune diseases [69]. An alternative explanation could be that individuals consuming more fruit and vegetables are more likely to adhere to a healthy diet and lifestyle, like engaging in regular physical activity, which has been shown to have a protective effect on RA risk [70].

The inverse non-linear association between vegetable intake and RA risk may be influenced by preparation methods and nutrient composition. Despite vegetables generally appearing to be beneficial, some studies indicate that cooked vegetables may have negative health effects due to harmful components such as acrylamide, advanced glycation end-products, and a loss of water-soluble vitamins. A recent study involving 400,000 adults found that raw vegetable intake was inversely associated with both cardiovascular disease incidence and mortality, unlike cooked vegetables [71]. Moreover, vegetable oils high in omega-6 PUFAs may promote low-grade inflammation and oxidative stress, unlike fresh vegetables containing omega-3 PUFAs with anti-inflammatory properties [72]. Another explanation could be that BMI mediates the relationship between vegetable intake and RA risk. Since the inverse association between vegetable intake and RA risk was over-adjusted for BMI as a confounder, the potential effect of vegetables on RA risk was diminished. This suggests that obesity [73] may play a role in RA development. 

Regarding cereals, our study showed a 3% reduction in RA risk with each additional 30 g/d of intake. This inverse association is consistent with recent large-scale cohort studies, including one from the UK Biobank showing a protective effect of breakfast cereals [16], as well as studies from America and France which linked whole grains and MD components with a reduced risk [34,55]. Despite differences in cereal types between the included studies, the low heterogeneity and robust results were confirmed by sensitivity analyses. The protective mechanism by which cereals may reduce RA risk could be similar to their effects on cardiovascular disease and type 2 diabetes [74,75], possibly attributable to the anti-inflammatory properties of cereal phytochemicals, fibre, vitamins, and minerals [76].

Associations between alcohol consumption and a lower risk of RA have been reported in previous meta-analyses [13,77]. In our study, the risk estimate for total alcohol was 0.76, slightly lower than earlier estimates of 0.86. This difference could be attributed to our stricter inclusion criteria, as we only considered prospective cohort studies, which are less prone to bias compared to prior meta-analyses that included both retrospective and prospective data [13,78]. This study includes more recent data, featuring over four times the number of RA cases and an average follow-up duration of 22.5 years, providing more robust evidence. Furthermore, this inverse association was consistently seen in the analysis of seropositive RA. Alcohol may exert a protective effect, particularly in individuals with autoantibodies. A Swedish population-based study has shown that low-to-moderate alcohol consumption may reduce the risk of developing ACPA-positive RA by mitigating the combined effects of smoking and the human leucocyte antigen (HLA)-DRB1 shared epitope [78]. Among alcoholic beverages, beer intake was negatively associated with RA risk, potentially contributing to the overall protective effect of total alcohol. A possible mechanism involves polyphenols like resveratrol, which may modulate the immune response by inhibiting pro-inflammatory cytokines [79], altering immune cell activity [80], and reducing oxidative stress through free-radical neutralisation [81]. Further intervention studies are warranted to clarify this beverage-specific relationship. 

The dose–response analysis revealed a J-shaped relationship, suggesting that low-to-moderate alcohol consumption is relatively safe, with benefits diminishing beyond 60 g/week. These findings could be used to refine the alcohol consumption guidelines, especially for individuals with rheumatic and musculoskeletal diseases, by identifying the optimal level of consumption for maximum benefit. We observed this inverse association between alcohol consumption and RA risk in studies from both America and Europe, but not in the limited data from Asia. This discrepancy may result from insufficient statistical power in the Asian studies or differences in the types of alcohol commonly consumed in Asia, which may influence the association with RA risk. The subgroup analysis indicated that the protective effect was consistent across both female and mixed-sex groups, suggesting that the effect may be applicable broadly. Future studies with sex-specific analyses and more extensive geographic coverage could strengthen the evidence.

The relationship between tea consumption and RA risk appears to be complex and controversial. Two meta-analyses found no association [82,83], while some studies suggested an inverse relationship [15,54]. Our analysis identified, for the first time, a positive linear association between higher tea consumption and increased RA risk, particularly in individuals with over 10 years of follow-up. Unlike previous meta-analyses [78,79], we found no significant association of total coffee with RA risk in extreme intake comparisons. However, we did find a non-linear positive relationship for total coffee and an apparent significant association for caffeinated coffee, suggesting that higher intake may elevate RA risk. This may be due to the pro-inflammatory effect of caffeine when consumed in excess, despite its known anti-inflammatory properties in small amounts.

Caffeine, a common component of both tea and coffee, could play a role in influencing RA risk. Supporting our findings, evidence from animal studies has supported the possibility that caffeine combined with sucrose can exacerbate inflammation [84]. Moreover, tea and coffee consumption is often linked to Western diets high in refined sugars and unhealthy fats, which are known to promote inflammation [81]. The inconsistencies across studies may be due to variations in tea and coffee types, intake levels, and study duration. More well-controlled trials are needed to clarify these associations.

In contrast, no evidence was found of an association for decaffeinated coffee or sugar-sweetened soda. The lack of findings for decaffeinated coffee may be attributed to its lower caffeine content, which likely contributes to the association observed with caffeinated coffee. Interestingly, a positive association between decaffeinated coffee and increased risk was found in both linear dose–response and extreme intake comparison meta-analyses under the common-effects model but not the random-effects model. This discrepancy may be due to differences in study demographics or methodologies, which are more apparent in random-effects models. For soda, despite the well-documented link between excessive dietary sugar intake and inflammation [85], the absence of significant findings may reflect heterogeneity across studies or differences in soda formulations. Further research is needed to investigate the specific components of these beverages and their interactions with other dietary factors. 

Our study found that oily fish and vitamin D supplements were associated with a reduced RA risk in the non-linear dose–response analysis, but no evidence of association was observed in either the comparison of extreme categories or the linear dose–response meta-analysis. Because of this inconsistency, there is a need for further research to provide more conclusive evidence.

This study has several limitations. First, although we focused on prospective cohort studies, observational studies are still prone to selection bias, reverse causality, publication bias, and small-study effects. Second, dietary assessment errors are unavoidable. Most studies relied on a single dietary measurement, which may not accurately reflect long-term intake. The use of different dietary assessment methods (e.g., FFQs, dietary records) and inconsistent intake units could also introduce heterogeneity. Under-reporting of alcohol consumption and changes in dietary habits over time may further weaken the observed associations with RA risk.

Third, unmeasured or residual confounding cannot be avoided. Low-to-moderate alcohol drinkers may have other beneficial lifestyle factors or a higher socioeconomic status compared to non-drinkers or heavy drinkers. For example, a cross-sectional study of the American population indicated that alcohol consumers tend to exercise more vigorously [86]. Additionally, dietary nutrients are consumed as part of an overall diet, making the total effect potentially greater than the sum of the individual components, and interactions between nutrients should be considered. Finally, our findings may not be fully generalisable to low- or middle-income countries due to differences in diets, food availability, or to males, as most participants were female and from America and Europe.

Our results indicate significant relative risk reductions for RA associated with higher consumption of total alcohol and certain food groups. However, a limitation is the absence of data on adjusted absolute risk differences, which restricts our ability to estimate the actual number of RA cases that could be prevented by dietary changes. Most of the included studies reported relative risks without providing baseline incidence rates or adjusted absolute risks. Future research should prioritise the reporting of both relative and absolute measures of risk, enabling a more comprehensive assessment of the public health impact of dietary factors on RA risk. Furthermore, while dietary factors may influence the severity of RA, this study did not address that aspect, leaving the role of diet in disease progression as an important area for future investigation.

This meta-analysis has some strengths. It included a large number of participants and RA cases, allowing for well-powered analyses of 32 dietary factors in relation to RA risk. This meta-analysis included only prospective cohort studies, which are less prone to recall bias compared to case–control studies. Dose–response analyses enabled exploration of the direction of the relationships and possible threshold effects, as well as an assessment of the credibility and reliability of the evidence.

## 5. Conclusions

This meta-analysis identified that total alcohol, fruit, cereals, oily fish, and vitamin D supplements were associated with a reduced risk of RA, supported by evidence of low-to-moderate credibility. Tea and coffee were associated with an increased risk. There is no evidence of an association between sugar-sweetened soda and RA risk. Further research is needed to confirm these associations and explore whether specific dietary patterns or nutrients could emerge as a viable strategy for RA prevention.

## Figures and Tables

**Figure 1 nutrients-16-04050-f001:**
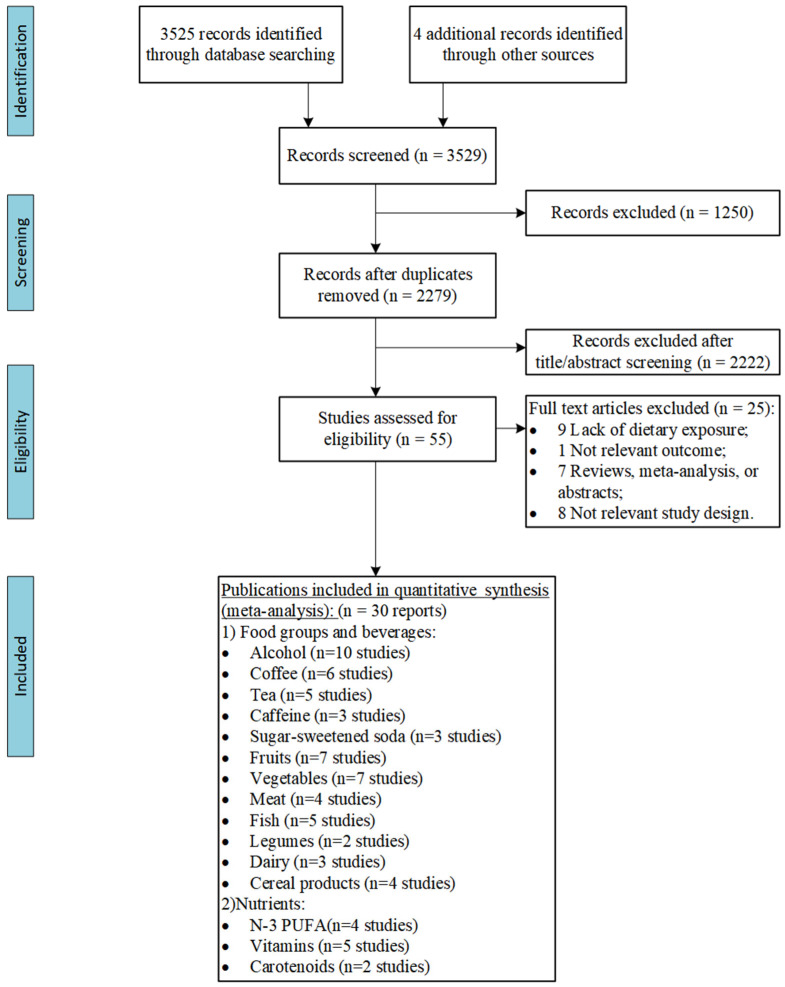
PRISMA flow chart summarising article retrieval and screening process.

**Figure 2 nutrients-16-04050-f002:**
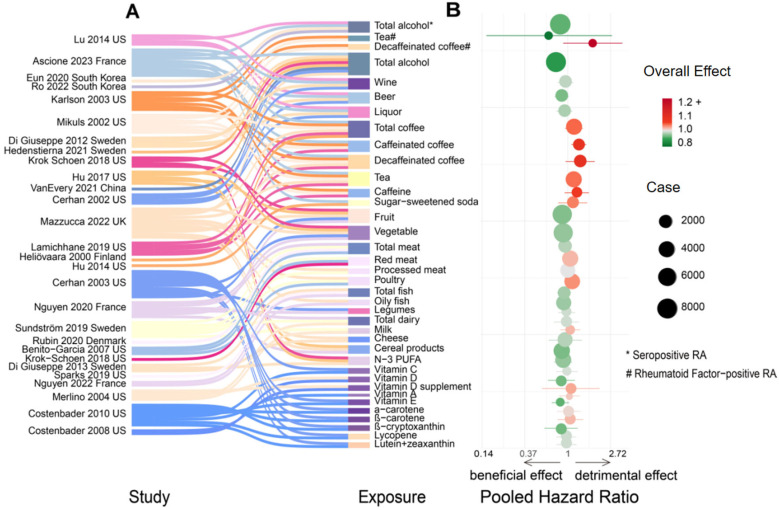
The summary of pooled HRs (95% CIs) for different dietary exposures and incident RA comparing the highest with the lowest categories. (**A**) Sankey diagram illustrating the 21 food groups, beverages, and 11 nutrients across 28 included cohort studies. (**B**) OncoPrint chart displaying the pooled hazard ratios (HRs) for incident RA, incident seropositive RA, and incident RF-positive RA, according to 37 food components and nutrients. Green colour represents a beneficial effect on RA incidence; red colour represents a detrimental effect on RA incidence. This figure does not indicate statistically significant results.

**Figure 3 nutrients-16-04050-f003:**
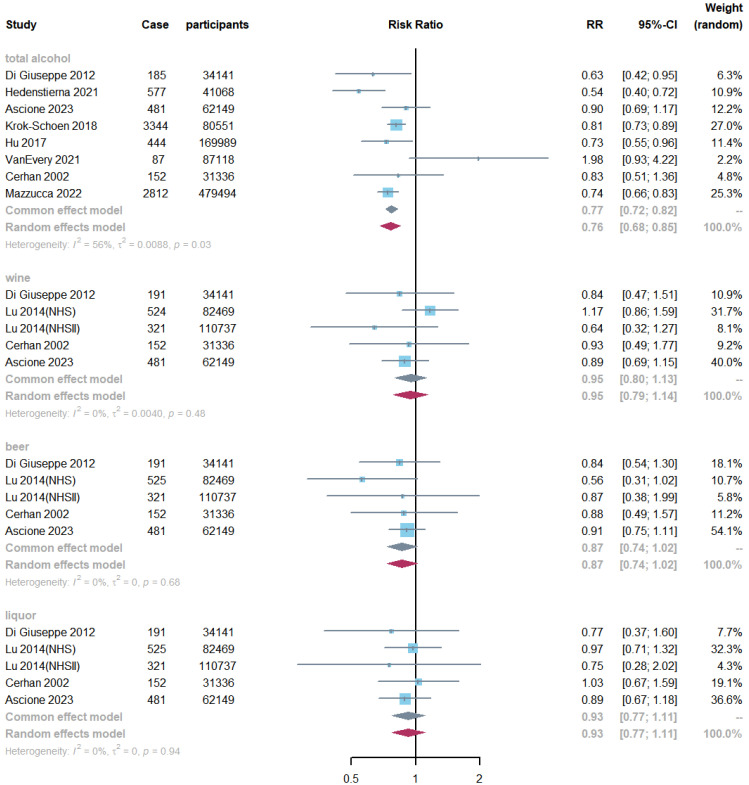
Meta-analysis of alcohol consumption and risk of RA comparing the highest with the lowest categories. Diamonds represent pooled estimates from random effects analysis.

**Figure 4 nutrients-16-04050-f004:**
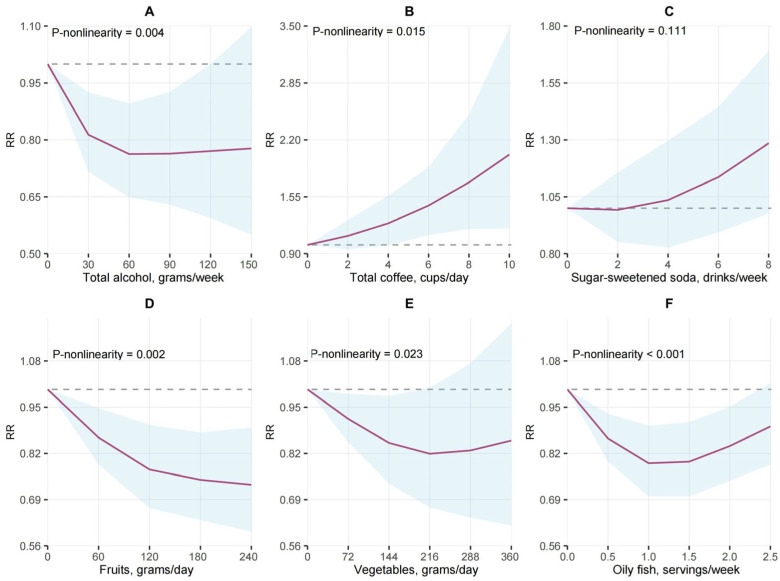
Non-linear dose–response association of food component consumption with risk of RA using restricted cubic splines. Relations for food components of (**A**) total alcohol, (**B**) total coffee, (**C**) sugar-sweetened soda, (**D**) fruits, (**E**) vegetables, and (**F**) oily fish. Solid lines represent the fitted non-linear trend and blue shade represents the pointwise 95% CIs.

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
