# Peer review of "Dose–Response Associations Between Diet and Risk of Rheumatoid Arthritis: A Meta-Analysis of Prospective Cohort Studies"

_nutrients, 2024, doi:10.3390/nu16234050_

Round 1

Reviewer 1 Report

Comments and Suggestions for Authors

Dear authors,

you conducted a meta-analysis of 30 studies on diet and the risk of developing rheumatoid artritis.  The majority only relate to females. Only 3 studies were conducted in Asia, the majority in America and Europe. 

The elaborations on nutrition are comprehensive, but the information on the underlying disease examined, rheumatoid arthritis, is limited. Were illnesses reported or confirmed by the investigators evaluated? Did the classification criteria have to be met? The reported number of seropositive patients is higher than expected. Only the relative risk was reported, but never the incidence described in the study. 

In the introduction it is written that RA is characterized by autoantibodies. This only applies to 2/3 of patients and is ultimately dispensable as a statement as it will not be discussed below. 

The statement about irreversible disability and increased mortality is only true for untreated diseases. The majority of patients no longer have any expected disability or increased mortality under (guideline-based) therapy.

The statement “diet could influence the development of RA” should be put into perspective. The pathogenesis is multifactorial and none of the diet data can explain the higher incidence in women and older people.

Overall, the relative influences of diet are also small. 

Unfortunately, the study cannot provide any information about the severity of the disease; dietary influences may influence the severity rather than the manifestation at all. 

Some data appears contradictory. Figure 2 shows an unfavorable effect of tea and vitamin D supplementation (ret dot), but not vitamin D (what's the difference?). The text states (lines 288-289) that vitamin D substitution reduces the risk by 20% and tea consumption is associated with a negative risk ratio (line 320). In line 382 it is again a positive association for tea. 

The Asian studies show a very unfavorable effect of alcohol consumption, which is not sufficiently discussed.

Furthermore, the often clear gender differences in the supplement are not adequately discussed, as there is a clear dependence on gender in Europe and America, especially when it comes to alcohol consumption and meat.

Other variables associated with increased risk such as BMI, exercise and nicotine consumption were not consistently adjusted (or were not described). This was also discussed, but should be discussed more obviously as a relevant bias.

Figure 2 also contains the editorial note in the explanation as to where tables should be placed in the text. In line 273 there is a missing space after "cheese".

Reviewer 2 Report

Comments and Suggestions for Authors

Please see enclosed pdf

Reviewer 3 Report

Comments and Suggestions for Authors

The article is very well written and the analysis of the result is perfect. These are very few minor questions regarding the context of the article.

  1. Could you please mention how the dose of the diet has been calculated, as the objective of the article is based on the dose-response association of diet and RA
  2. What does the RA identification mean—the presence or absence of RA?
  3. Line 257: Could you please specify the line “subgroup analysis”
  4. Could you please specify the meaning of moderate/lower quality evidence
  5. Could you please specify the meaning of highest with the lowest categories

As already mentioned in the discussion, the direct association of a single factor (alcohol intake/vegetables/fruits) to RA cannot be directly related as multiple factors can come into play. However, a supporting literature on each factor’s mechanism on the RA effect can rationalize the conclusions from the meta-analysis. For instance, diets rich in omega 6 and gluten may increase the risk of RA as they are associated with increased inflammation. Hence vegetables rich in omega 6 may increase the RA risk while vegetables rich in omega 3 may decrease the risk of RA.

Reviewer 4 Report

Comments and Suggestions for Authors

This systematic review addressed the question of whether various dietary factors influence the risk of developing RA, with a focus on food groups such as alcohol, fruits, vegetables, cereals, caffeinated coffee, and tea. The study included prospective cohort studies published between 2000 and 2024, analyzing their findings on RA risk and dietary intake. The review appears to have followed appropriate inclusion criteria, focusing on studies that reported relative risks (RRs) with confidence intervals (CIs) for RA incidence. . The following aspects should be addressed:

- What tools or criteria were used to assess the methodological quality of the included studies, and were they appropriate for evaluating observational studies?

- Were standardized methods used to assess dietary exposures across studies, considering the variability in dietary reporting?

- Were subgroup analyses performed to investigate potential sources of heterogeneity, such as geographic location, dietary measurement methods, or participant characteristics (e.g., age, gender, lifestyle factors)?

- How might variations in RA diagnosis (seropositive vs. seronegative) influence the association with dietary factors?

- What are the effects of non-linear dose-response relationships for dietary components other than alcohol and vitamin D?

- Were potential confounding factors (e.g., physical activity, genetic predisposition, or socioeconomic status) adequately controlled, and could residual confounding affect the validity of the results?

- Were time-varying dietary exposures considered to account for changes in eating habits over time, and could more robust longitudinal data improve the accuracy of the associations?

Round 2

Reviewer 2 Report

Comments and Suggestions for Authors

The manuscript has been improved